Corrected: Publisher correction

# Ballistic geometric resistance resonances in a single surface of a topological insulator

Hubert Maier[1], Johannes Ziegler[1], Ralf Fischer[1], Dmitriy Kozlov[2,3], Ze Don Kvon[2,3], Nikolay Mikhailov[2], Sergey A. Dvoretsky[2] & Dieter Weiss[1]

Transport in topological matter has shown a variety of novel phenomena over the past decade. Although numerous transport studies have been conducted on three-dimensional topological insulators (TIs), study of ballistic motion and thus exploration of potential landscapes on a hundred nanometer scale is for the prevalent TI materials almost impossible due to their low carrier mobility. Therefore, it is unknown whether helical Dirac electrons in TIs, bound to interfaces between topologically distinct materials, can be manipulated on the nanometer scale by local gates or locally etched regions. Here we impose a submicron periodic potential onto a single surface of Dirac electrons in high-mobility strained mercury telluride (HgTe), which is a strong TI. Pronounced geometric resistance resonances constitute the clear-cut observation of a ballistic effect in three-dimensional TIs.

[1] Institute of Experimental and Applied Physics, University of Regensburg, 93053 Regensburg, Germany. [2] A.V. Rzhanov Institute of Semiconductor Physics, 630090 Novosibirsk, Russia. [3] Novosibirsk State University, 630090 Novosibirsk, Russia. Correspondence and requests for materials should be addressed to D.W. (email: dieter.weiss@ur.de)

Topological insulators that feature conducting surface states while the bulk is insulating are a new class of materials with unique physical properties (see, e.g., refs. [1,2] and references therein). By combining materials with different topological index, e.g., HgTe which has an inverted band structure with HgCdTe (or CdTe) being a conventional semiconductor, helical two-dimensional electron systems (2DES) form at the interface. These states have a linear, Dirac-like electron dispersion $E(\mathbf{k})$ and are protected from direct backscattering by time reversal symmetry[1,2]. The 2DES of surface states encases the (ideally) insulating bulk. The electron spin in theses surface states is helical, i.e., locked to the electron's $\mathbf{k}$ vector and spin-up ($\uparrow$) and spin-down states ($\downarrow$) are for systems with time reversal symmetry connected by Kramers degeneracy, $E(\mathbf{k}, \uparrow) = E(-\mathbf{k}, \downarrow)$. Thus, any $\mathbf{k}$-space state is only singly occupied. This means that the connection between carrier density $n_s$ and Fermi wave vector $k_F$ is given by $k_F = \sqrt{4\pi n_s}$ and not $k_F = \sqrt{2\pi n_s}$, as in a conventional 2DES. Clear signatures of the unusual spin texture of three-dimensional TIs have first been seen in $\mathrm{Bi}_{1-x}\mathrm{Sb}_x$ by angle-resolved photoemission spectroscopy[3,4].

Here we probe ballistic geometric resonances in an antidot lattice, which directly reflect the wave vector of helical Dirac fermions. An antidot lattice consists of a periodic array of holes etched into the surface and constitutes a periodic, repulsive potential for the electrons in the 2DES[5–7]. Pronounced resonances occur in the magnetoresistance when the ratio of cyclotron radius $R_c = \hbar k_F/(eB)$ ($\hbar$ = reduced Planck constant, $e$ = elementary charge, $B$ = magnetic field) and period $a$ of the antidot lattice enables stable orbits around one or a group of antidots[5], stabilized by the repulsive potential[8]. Similar effects

were predicted and observed in graphene, also a system with Dirac-like dispersion[9–11]. The $B$-field positions at which the resistance peaks appear are a direct measure of $k_F$. Such resonances have been used, e.g., to experimentally proof the concept of composite fermions[12] and the formation of Wigner crystals[13]. The observation of these commensurability effects requires that the electron mean free path $\ell_e \gg a$, i.e., high electron mobilities. Hence, strained HgTe which belongs to the class of strong TIs[14,15] is ideal for these experiments as it features very high electron mobilities of order $5 \times 10^5$ cm$^2$ (Vs)$^{-1}$, corresponding to electron mean free paths of around 5 microns. Experimental verification that strained HgTe is a TI has been provided before by transport[16,17], capacitance[18], and photocurrent experiments[19]. In this material, we observe clear-cut geometric resonances reflecting the superimposed lateral periodic potential.

## Results

**Antidot lattices and characteristic geometric resonances.** The present experiments were carried out on strained 80 nm-thick HgTe films, grown by molecular beam epitaxy on CdTe (013). For details, see ref. [19]. The cross section of the heterostructure is sketched in Fig. 1a. Square antidot lattices with periods $a = 408$, 600, and 800 nm were defined by electron beam lithography and etched through the cap layers slightly into the HgTe layer. In order to keep the high electron mobility of the surface electrons, we used wet chemical etching to engrave the antidots gently into the material. An electron micrograph of a corresponding antidot lattice is shown in Fig. 1b, illustrating that the antidots get only a few nm into the 80 nm-thick HgTe layer. Transport and

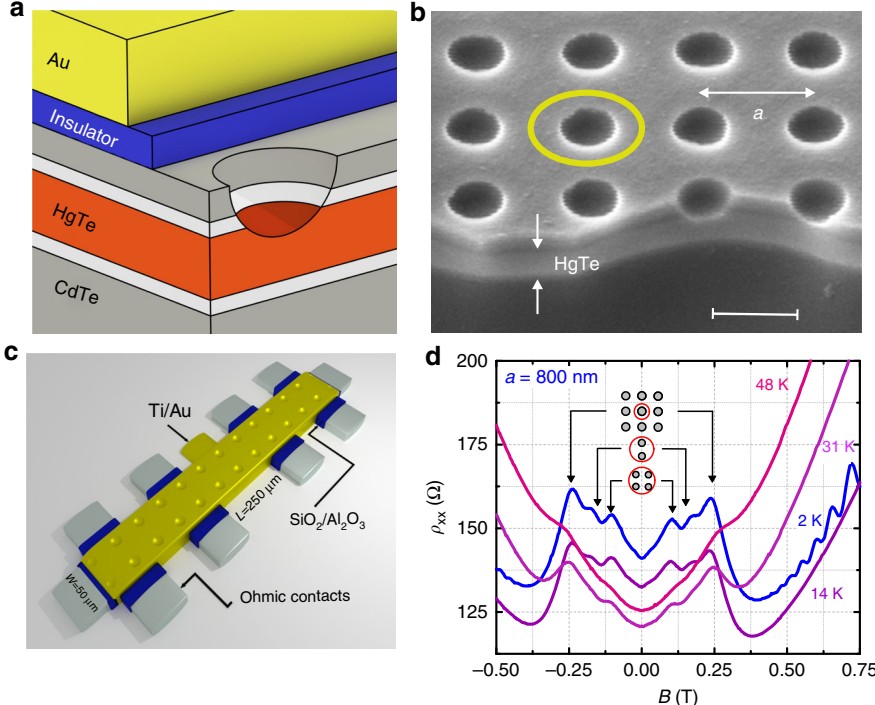

**Fig. 1** Sample layout and antidot resonances. **a** Layer sequence grown on a virtual CdTe (013) substrate having a 0.3% larger lattice constant than HgTe, causing tensile strain[19]. One antidot, slightly etched into the HgTe top layer is sketched. The whole structure is covered by a 30 nm/100 nm-thick SiO$_2$/Al$_2$O$_3$ insulator and a metallic top gate consisting of 10 nm/100 nm Ti/Au. **b** Scanning electron micrograph picture of an antidot lattice with a period of ~400 nm. The sample was tilted by 50° and the scale bar is 300 nm. The 80 nm-thick film of strained HgTe, covered by 20 nm HgCdTe and 40 nm CdTe cap layers stands out by the brighter contrast. **c** Schematic Hall bar geometry with dimensions. **d** $\rho_{xx}(B)$ of the 800 nm antidot lattice at different temperatures and $n_s^{top} \sim 1.7 \times 10^{15}$ m$^{-2}$. Peaks in the resistivity correspond to commensurate orbits around 1, 2, and 4 antidots as sketched in the inset. The corresponding arrow positions, characteristic for a strong antidot potential, have been calculated using $B_1 = \hbar k_F/(0.5ea)$, $B_2 = \hbar k_F/(0.8ea)$, and $B_4 = \hbar k_F/(1.14ea)$ with $k_F = \sqrt{4\pi n_s^{top}}$

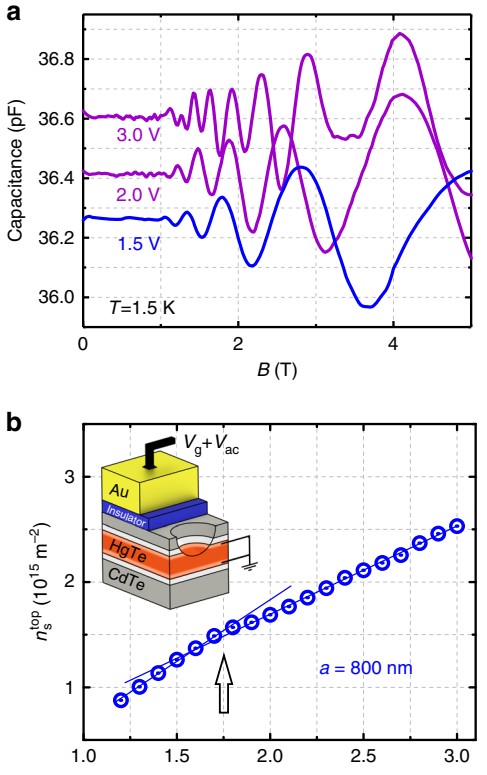

**Fig. 2** Capacitance oscillations and density. **a** Oscillations of the quantum capacitance with constant $\Delta(1/B)$ periodicity reflect Landau quantization of Dirac electrons on the top surface. The corresponding carrier density is given by $n_s^{top} = e/(2\pi\hbar\,\Delta(1/B))$. **b** Inset: Capacitance is measured by superimposing a small ac signal (0.01 V at 50 Hz) onto a dc voltage (which tunes $E_F$) and probing the ac current phase sensitively[18]. The blue points show the carrier density of the top surface, $n_s^{top}$, obtained from the periodicity of the capacitance oscillations. The arrow marks the gate voltage at which the slope $dn_s^{top}/dV_g$ changes. For $V_g < 1.75$ V the filling rate is $1.22 \times 10^{15}$ m$^{-2}$ V$^{-1}$, for $V_g > 1.75$ V it is $0.87 \times 10^{15}$ m$^{-2}$ V$^{-1}$

capacitance experiments were carried out on devices with Hall bar geometry, sketched in Fig. 1c. For transport measurements, carried out at temperatures $T$ between 1.5 and 48 K, we used standard lock-in techniques. Typical traces of resistivity $\rho_{xx}(B)$, taken at different temperatures, are shown in Fig. 1d and display clear antidot resonances. The positions of the resonances, marked by arrows, correspond to commensurate orbits around one, two, and four antidots occurring at $R_c/a = 0.5$, 0.8, and 1.14, respectively[5], shown in the inset of Fig. 1d. With increasing temperature, the Shubnikov-de Haas (SdH) oscillations fade away but the antidot resonances are still visible at higher $T$ indicating their semi-classical origin.

The occurrence of clear antidot resonances raises the question about the origin of the repulsive potential as the two-dimensional topological surface states are bound to the interface. Thus, by etching the holes, the boundary between HgTe and vacuum is locally shifted deeper into HgTe but there are no regions devoid of electrons as in a conventional 2DES-based antidot lattice. Opening the cap layer at the antidot positions and exposing the HgTe surface to ambient conditions most likely increases locally the $p$-doping thus constituting an electrostatic potential barrier between etched and pristine regions of the HgTe surface. Etching the array of holes might also introduce a periodic strain pattern, which contributes to the antidot potential in a similar way as ripples affect the band structure in Bi$_2$Te$_3$[20]. Although the

detailed microscopic origin and shape of the antidot potential is unknown yet, its effect on electron transport is clearly visible in Fig. 1d. From the similarity to geometric resonances in GaAs-based 2DES, we conclude that the antidot potential has the diameter of the etched holes and drops precipitously but softly toward the region between the antidots.

**Carrier densities and geometric resonances**. To establish the connection between Fermi wave vector $k_F$ and carrier density, we need accurate knowledge of the electron density $n_s^{top}$ on the top surface, i.e., the surface locally dented by the antidots. The period of SdH oscillations, taken at high $B$, or Drude transport data give the total carrier density $n_s$ of the TI system, i.e., the sum of densities on top and bottom surface and, for $E_F$ in the conduction band, of bulk electrons[17]. Therefore, we resort to measurements of the quantum capacitance which reflects the electronic density of states (DoS) and is sensitive to the top surface only[18]. In capacitance experiments, the top layer dominates, as it screens, to a large extent, the electric field from the gate. The capacitance is measured by adding a small ac voltage to the dc top gate voltage $V_g$ and probing the ac current phase sensitively. Corresponding data are shown in Fig. 2a. The capacitance oscillations of the 800 nm period antidot sample, shown for three different gate voltages, stem from Landau quantization, echoed in the measured DoS. From these measurements, we can extract the carrier density $n_s^{top}(V_g)$ of the top surface, shown as a function of the gate voltage in Fig. 2b. Here we have used that the spin-degeneracy factor $g_s$, utilized to extract the carrier density from the period of the capacitance oscillations, is 1, i.e., the Landau levels are not spin degenerate[18]. These data show a striking change of slope at ~1.75 V, marked by an arrow. This reflects a change in the filling rate $dn_s^{top}/dV_g$ and occurs when the Fermi level enters the conduction band[17]. This kink in slope is characteristic for quantities like low-$B$ SdH oscillations[17], quantum capacitance[18], or antidot peak position (see below), which all stem from a single topological surface.

**Gate voltage and antidot period dependencies**. The dc top gate voltage on our devices allows tuning the carrier density and thus the position of the Fermi level $E_F$. The characteristic evolution of the sheet resistance $\rho_\square = \rho_{xx}\,(B=0)$ with $V_g$ is shown for three antidot samples and an unpatterned reference sample in Fig. 3a. The maxima of $\rho_\square$, located between $V_g \sim 0.5$ and 1 V, reflect the charge neutrality point (CNP), corresponding to an $E_F$ position located slightly in the valence band (see band structure in Fig. 3b)[17]. The fact that the CNPs of the patterned samples are shifted to higher gate voltages (clearly for the 408 and 800 nm, only weakly for the 600 nm sample) implies local $p$-doping.

For gate voltages to the right of the CNP peak, $E_F$ starts to move into the gap. The dark blue traces in Fig. 3c show $\rho_{xx}(B)$ data for $V_g$ between 1.2 and 1.7 V at which $E_F$ is in the gap. The evolution of the fundamental antidot peak position $B_1$ for different $V_g$, matches perfectly the calculated position at

$$B_1 = \frac{2\hbar\sqrt{4\pi n_s^{top}}}{ea}, \qquad (1)$$

marked by arrows. Here for each particular gate voltage $n_s^{top}$ has been derived from the oscillation period of the quantum capacitance. The antidot peaks even withstand moving $E_F$ into the conduction band (purple curves in Fig. 3c), showing that bulk electrons do not contribute. This happens for $V_g > 1.75$ V, as deduced from the kink in the capacitance data of Fig. 2b. Although current also flows through the bottom surface and, for $V_g > 1.75$ V, through bulk regions, thus contributing to the resistivity, only electrons in the top layer contribute to the geometric transport resonances. At lower bias, however, where $E_F$ is in the valence band (light blue traces) the antidot peaks quickly

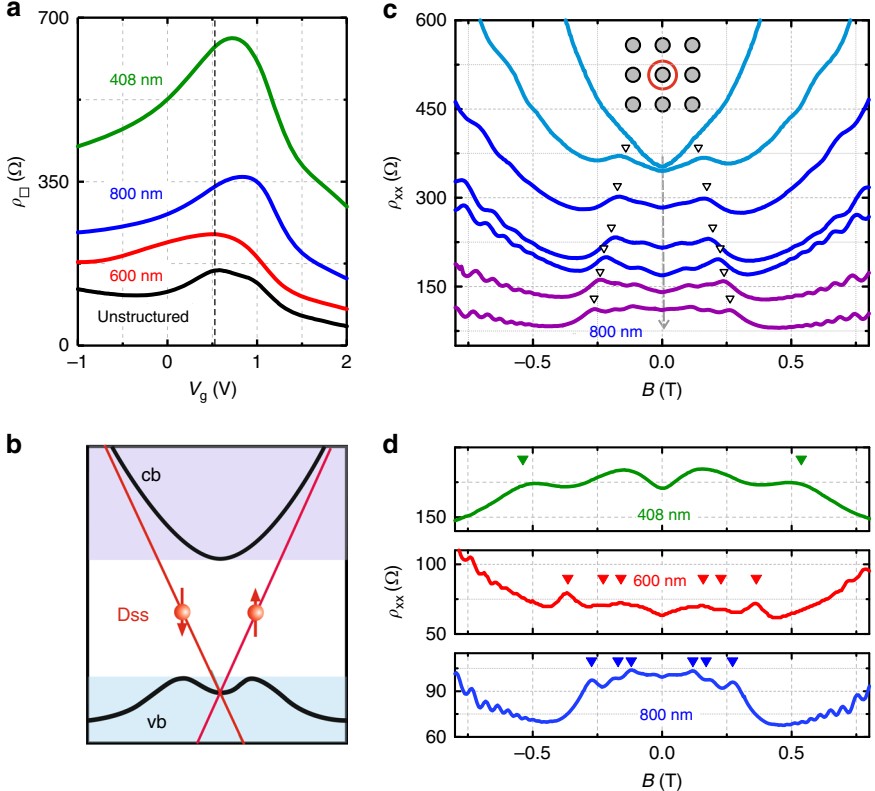

**Fig. 3** Magnetoresistance of three antidot superlattices. **a** Sheet resistance $\rho_\square$ vs. $V_g$ of an unpatterned reference sample and three antidot lattices with $a = 408$, 600, and 800 nm. The dashed line marks the CNP of the unpatterned sample. **b** Schematic band structure showing the conduction band (cb), the valence band (vb), and the Dirac surface states (Dss). **c** $\rho_{xx}(B)$ traces taken from the 800 nm antidot lattice at gate voltages $V_g = 0.8$, 1, 1.2, 1.4, 1.7, 2, 2.4 V (from top to bottom). The dashed arrow points in the direction of increasing $E_F$. The fundamental antidot peaks, corresponding to a cyclotron orbit fitting around one antidot (inset) have been calculated from $n_s^{top}$ and are marked by arrows. **d** Low field $\rho_{xx}(B)$ traces showing antidot resonance for the three investigated periods. $n_s^{top}$ was adjusted to ~2.2 × 10$^{15}$ m$^{-2}$ for all traces via $V_g$. The arrows mark calculated values of the field $B_1$, and for the 600 and 800 nm sample also $B_2$ and $B_4$. The lower zero field resistance of the 600 nm sample we ascribe to a longer exposure to 80 °C during insulator deposition

disappear and are hardly visible for $V_g = 1$ V, i.e., close to the CNP. After we have shown the evolution of the antidot peak position as a function of $V_g$, i.e., carrier density, we present in Fig. 3d the fundamental peak positions for three different periods at constant carrier density. Also, here the position of the fundamental peak is given by $B_1$ (Eq. 1) and marked by arrows in Fig. 3d. For the 600 and the 800 nm device also the position of the orbits around two ($B_2$) and four antidots ($B_4$) are shown. For calculating the resonance position, we have used $k_F = \sqrt{4\pi n_s^{top}}$, i.e., the relation between Fermi wave vector and carrier density of a two-dimensional electron gas which is not spin degenerate (see inset of Fig. 4b). The nearly perfect agreement between measured and calculated resonance position is in line with antidot resonances stemming from helical Dirac surface states, which are spin-polarized, as sketched in Fig. 3b, and probe a single topological surface. Figure 4 provides further evidence that the antidot peaks sample the properties of the top surface only. For that, we extract the carrier density from the $B$-field position at which the fundamental resonance occurs, using $n_s^{top} = (eaB_1)^2/(16\pi\hbar^2)$. The result is shown by the red circles in Fig. 4a. These data points resemble closely $n_s^{top}$, derived from the quantum capacitance oscillations in Fig. 2b, also shown in Fig. 4a. Most importantly, $n_s^{top}(V_g)$, acquired from the antidot resonance position displays a change of slope at about 1.75 V, in analogy to the capacitance data. This is a clear indicator that the antidot signals root in a single topological surface. Therefore, antidot resonances can be used to extract the carrier density of the top surface also at higher temperatures at which quantum oscillations are smeared out. In contrast, the total carrier density,

determined here from the position of the quantum Hall plateaus at high fields and shown in the upper inset of Fig. 4a echoes the total carrier density, i.e., the sum of $n_s^{top}$, the carrier density of the bottom surface, $n_s^{bot}$, and, for $V_g > 1.75$ V, $n^{bulk}$ the bulk carrier density. The slope of the total carrier density $n_s(V_g)$, in contrast, is constant and reflects the constant filling rate $dn_s/dV_g$ (upper inset Fig. 4a).

Figure 4b shows the magnetic field positions $B_1$ of the fundamental antidot resonance occurring at $2R_c = a$ for all three investigated devices. The carrier density $n_s^{top}$ for each gate voltage has been taken from the quantum capacitance oscillations. The solid lines have been calculated using Eq. (1). The lines describe the data for the 600 and 800 nm antidot lattice perfectly well; in the case of the 408 nm lattice, there is a small deviation of ~10%. There, the capacitance was measured on a sample area where only a fraction of ~18% was covered with antidots. Therefore, the extracted $n_s^{top}$ is an average value, which is slightly higher than the carrier density in the antidot lattice. By assuming a 20% reduced carrier density in the antidot patterned areas compared to pristine ones, a plausible value for our system, we obtain a perfect match with experiment (green dashed line in Fig. 4b).

## Discussion

In the present work, we have imposed a (periodic) potential on the nanometer scale onto a single surface of Dirac electrons and observe clear geometric resonances, which stem from the upper topological surface. This is of central importance for all kinds of (ballistic) manipulation of helical Dirac electrons on the

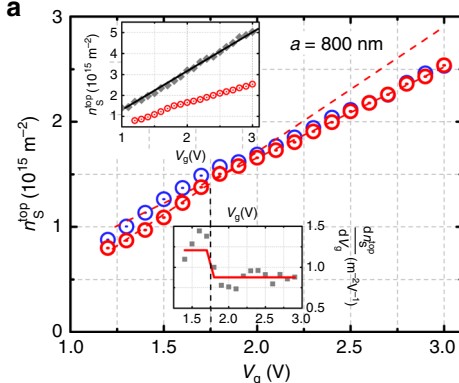

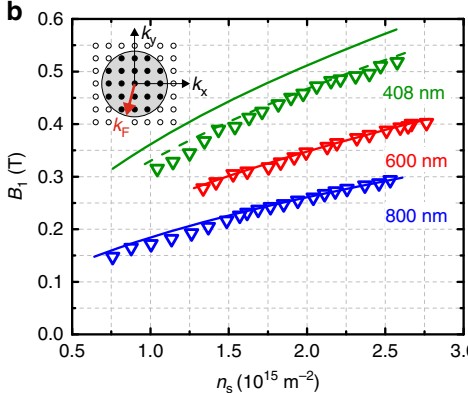

**Fig. 4** Geometric antidot resonances. **a** Carrier density $n_s^{top}$ extracted from the position of the fundamental antidot peak. $n_s^{top}(V_g)$ displays a distinct change of slope at ~1.75 V highlighted in the lower inset, which shows $dn_s^{top}/dV_g$ vs. $V_g$. Gray points in the inset are obtained by taking the average slope between three succeeding points, the red line is the best fit (dashed lines in main graph) of the slope above and below $V_g \sim 1.75$ V. The filling rate $dn_s^{top}/dV_g$ in the gap is $1.21 \times 10^{15}$ m$^{-2}$ V$^{-1}$ and $0.84 \times 10^{15}$ m$^{-2}$ V$^{-1}$ for $E_F$ in the conduction band. These values are very close to the ones obtained from capacitance oscillations, displayed as blue circles for comparison. The upper inset shows the total carrier density $n_s$ extracted from the magnetic field position of the quantum Hall plateaus compared to the red data points of the main graph (same units as main graph). **b** $B$-field position of the fundamental antidot resonance of the three different antidot lattices compared to the expected position (solid lines) using Eq. (1). The dashed line for the 408 nm sample shows the calculated $B_1(n_s)$ trace with corrected carrier density (see text). The inset shows a schematic of **k**-space in which the occupied states (black dots) are not spin degenerate

nanoscale. The antidot resonances probed here reveal the Fermi wave vector, being consistent with Dirac surface states, which are singly occupied in **k**-space.

## Methods

**Device fabrication**. The first step in device fabrication is defining the antidots by electron beam lithography and etching the antidot slightly into the HgTe layer by using a wet chemical etching solution containing 0.1 ml bromine, 100 ml ethylene glycol, and 25 ml distilled water. To ensure reproducible etching rates, the solution is cooled to 0 °C. The etched antidots have a diameter between 170 and 190 nm. After etching the antidots, we defined and etched a Hall bar by optical lithography and wet chemical etching. By etching the Hall bar after nanostructuring, we remove differently deep etched antidots, which occur along the perimeter of the patterned area between the potential probes so that they do not matter in transport experiments. As gate insulator, we deposited 30 nm SiO$_2$ using a plasma-enhanced chemical vapor deposition process followed by atomic layer depositing of 100 nm Al$_2$O$_3$; both processes were carried out at 80 °C. In a final patterning step, we

defined a titanium/gold (10 nm/100 nm) gate by thermal evaporation. Finally, ohmic contacts to HgTe where formed by indium soldering.

**Capacitance measurements**. The capacitance measurements where performed at 1.5 K in a $^4$He cryostat with magnetic fields up to 14 T. For these measurements, we used an Andeen–Hagerling (AH) 2700A capacitance bridge with a modulation amplitude of 0.01 V at 50 Hz.

**Magnetotransport measurements**. We used conventional Hall bar geometries having a width of 50 μm and a length of 650 μm with four potential probes of width 5 μm on each side, separated by either 100 or 250 μm. The resistivity $\rho_{xx}(B)$ was extracted from areas of size 100 μm × 50 μm or 250 μm × 50 μm, respectively. Standard four-terminal measurements were carried out at 1.5 K, unless specified otherwise, using digital lock-in amplifiers (Signal Recovery 7265/EG&G Instruments 7260) at a frequency of 13 Hz. A Yokogawa 7651 DC source was used to apply the gate voltage.

**Data availability**. The data supporting the findings of this study are available within the paper and from the corresponding author on request.

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

## Acknowledgements

We thank Klaus Richter for valuable discussions and acknowledge funding by the German Science Foundation (DFG) via SPP 1666 and by the Elite Network of Bavaria (K-NW-2013-258). D.K. acknowledges support from RF President Grant No. MK-3603.2017.2, and N.M. from RFBR Grant No. 15-52-16008.

## Author contributions

D.W. conceived the experiments. N.M. and S.A.D. grew the material. H.M. prepared the sample, performed the measurements, and analyzed the data with contributions from D.K., Z.D.K., J.Z., R.F., and D.W. All authors contributed to analyzing and interpreting the data. H.M. and D.W. wrote the manuscript.

## Additional information

**Competing interests:** The authors declare no competing financial interests.

