## [Peer Review File · Nature Communications]

Reviewers' comments:

Reviewer #1 (Remarks to the Author):

The authors studied the transport of ballistic Dirac Fermions in the topological insulators surface states of strained HgTe films where the surface has been engineered to manipulate the motion of charges. An antidot lattice that modifies the semi-classical trajectories of the electrons was defined, a signature of which can be observed in the magnetoresistance measurements where resonances appear each time that the cyclotron radius has some specific values related to the lattice constant of the square antidote lattice.

The experiments have been carried out in a thorough and proper way. Results are convincing and support the conclusions of this work. The authors are able to extract the energy dependence of the Fermi wave vector and the results are in very good agreement with previous results obtained with different experimental techniques. The authors claim that the interest of their work lies in the fact that these experiments allow to directly access the value of the Fermi wave vector and that it demonstrates the ballistic nature of the motion of charges in HgTe 3D topological insulator.

I recommend this work for publication provided that the authors answer the concerns and remarks addressed below. The results are novel and the demonstration of the possibility to engineer topological surface states would have a clear impact for building electronic devices out of ballistic 3D topological insulators.

Remarks and concerns:

i) The authors show that their experiments directly probe the value of the Fermi wave vector. Nevertheless, the interest of this method with respect to other measurement techniques that also probe k_F is not clearly explained in the manuscript. For instance, it remains unclear what would be the advantage over Shubnikov de Haas oscillations (SdHO) measurements (at low and high magnetic field to extract all densities according to some previous work of the authors) of either the conductance or the quantum capacitance and even over angle resolved photo emission spectroscopy experiments. Indeed, SdHO also probe k_F directly (and not the density of charge/state) as well as ARPES measurements. According to my understanding, the strongest conclusion of this work is the observation of ballistic motion and the manipulation of the charge orbitals based on the engineering of the surface state in strained HgTe thin film.

ii) To my understanding, measurements of the quantum capacitance give a direct access to the density of states and therefore to the electronic density. Nevertheless, the electronic density is given in this work by the magneto oscillations of the quantum capacitance which are related to k_F instead of the density. Therefore, it is not surprising that k_F is found to correspond to a non-degenerate surface state since both methods are directly related to k_F . In order to have a strong conclusion about the spin density, the authors should compare the k_F extracted from the measurement of the antidot lattice with the one obtained from direct measurement of the density, i.e. from quantum capacitance measurements. If this would not be possible, they should not claim that the comparison of the k_F between both methods is a proof of the non-degeneracy of the surface states.

iii) For the 400nm, a correction accounting for the difference between the area of the antidot lattice and the total area would clearly strengthen the conclusion of this work. Three different lattice constants make the statistics strong enough as long as all of them are in good agreement with the theory. Even if the argument of the authors appears very reasonable, a quantitative estimation of the effect of the difference of areas would make their conclusions even more convincing.

iv) Even if the three different lattice constants are very convincing, it would be worth to realize experiments on different lattices (for instance a triangular lattice or a lattice with two different lattice constant) in order to demonstrate the relevance of antidot lattices for the manipulation of ballistic Dirac Fermions. Nevertheless, this should not be a requirement for publication but a suggestion to make the conclusion of this work even more interesting.

v) The authors did not discuss the nature of the perturbation induced by the antidot (weak or a strong perturbation). Strong or weak perturbation refers here either to dots which are completely depleted (Ensslin et al. Phys. Rev. B 41, 12307, Weiss et al. Phys. Rev. Lett. 66, 2790, Kang et al.

Phys. Rev. Lett. 71, 3850) or to a situation where the potential varies only weakly in the dot (Weiss et al. EPL 8, 179, Lorke et al. Phys. Rev. B 44, 3447, Deng et al. Phys. Rev. Lett. 117, 096601). One might naively expect a weak perturbation but the authors did not discuss this point in the manuscript. The work Lorke et al. (Phys. Rev. B 44, 3447) which justifies the interpretation of the data as done in the manuscript, should be cited here. Even if I imagine that the answer is probably negative, a relevant question would be: is there any experimental way to differentiate both scenarios?

vi) The inset in the figure 4.b is not clear. In general, it would be good to have the title of the different axis (inset included)

vii) In the introduction, it is very difficult to make the link between Majorana, spintronic and the present experiment. A more detailed description would be helpful for the reader to understand the interest of this work regarding those two fields.

viii) Do you have any explanation for the non-monotonous change of the resistance between 400 nm, 600 nm and 800 nm samples for the resistance? Is it due to the geometry of the sample? More generally, the exact geometry of the Hall bars are not indicated (length, width, width of the contact, number of contact, etc...). The schematic in the Fig. 1 is a nice drawing that makes clear how the samples are measured but it is not sufficient to get all useful information.

Reviewer #2 (Remarks to the Author):

The authors have created submicron periodic potentials onto the top surface of strained mercury telluride and they have observed pronounced geometric resistance resonances when the Fermi wave vector of the TI matches the size of the antidot arrays.

I think the present work is of high quality and it might be of great interest to others in the community and the wider field. The experimental data supports the claim of ballistic effects in 3D-TIs and there is enough evidence from the 3 different samples mentioned in the manuscript. Therefore, I would strongly recommend the paper for publication in Nature Communications. I only have minor comments.

1. Experiments and theories on antidot superlattices in graphene has been performed before, and they should be cited in this manuscript. Such examples: Pedersen et al. and T. Shen et al.
2. A brief explanation why the capacitance measurements only gives the top-surface carrier density should be included in the paper, since it is important for the main conclusions in the paper.

Reviewer #3 (Remarks to the Author):

I reviewed the paper "Ballistic geometric resistance in a single surface of a topological insulator" by Maier et al. The paper presents an experiment in which a periodic array of anti dots is patterned on the surface of a 3DTI by etching. The authors present a set of transport measurements in which resonances of the magnetoresistance are identified and from which the Fermi wave vector is extracted.

The setup is borrowed from an old set of experiments done on 2DEGs that targeted, for example, the physics of the half filled Landau level and from which one could extract the Fermi wave vector of composite fermions. Of course there, actual holes were punched into the 2DEG. Here, the authors use the unique property of the surface of the TI, which is that it sticks to any geometry and cannot be interrupted by non-magnetic perturbations. Hence, the surface state follows the bumps and curves of the etched device, unlike in the standard 2DEG.

The particles seem to avoid the antidots in the periodic array due to an electrostatic potential, although in principle the surface state does not terminate there. The authors state that the precise mechanism by which this repulsion is generated is not known but it is likely to form due to

different doping level created in these regions. The data clearly shows resonances in the expected positions so in this sense it seems to agree with the mechanism described in the manuscript.

Overall I find the manuscript of interest and the data convincing. Patterning of TI materials is an outstanding problem and has proven challenging in the past. It did not work well with magnets, for example. I think this design could prove useful if adapted in other cases as well, such for the anomalous Hall insulator, in which the transition into the topological phase crucially depends on the thickness of the TI. Therefore a proof of concept for patterning in TI is interesting and important. I am generally positive about this paper being published in Nature Communications. Before it is published I would like the authors to consider some points for discussion.

I think the authors should discuss in more detail how the electrostatic potential is formed and affects the structure and the effective length scales. If the particles avoid the antidot due to electrostatic repulsion, I would expect that the potential is not hard-wall like but soft and smooth in contrast with the drilled holes in 2DEGs. Is there an indication for that in any of the measurements? What is the effective size of an antidot? This is not a point like object as is seen clearly from fig1, its size is comparable to the distance between antidots.

In this context, I would also like to point out a related experimental study (see <https://arxiv.org/abs/1207.4468>) where periodic buckling was introduced in Bi₂Te₃ and resulted in a one-dimensional periodic potential. Among other things, in this paper, the authors discuss the effects of doping vs potential ripples from lattice deformations. It would be good if the authors could compare and contrast the physics presented in this paper with the one that emerges from the antidot array.

Finally, I would like to understand if the Berry phase of the Dirac particles could be directly probed by this experiment. Since we are discussing transport in the semiclassical regime, the semiclassical equations of motion should in principle contain such a contribution.

Response to Referees

We thank the Referees for the careful reading of our manuscript and very much appreciate the detailed and positive evaluation. Below we address each of the issues raised by the Reviewers point by point. Apart from the changes initiated by the Reviewers' comments and questions, we have also made a few changes to comply with the Nature Communications style. All changes are highlighted in the manuscript.

Dieter Weiss (on behalf of all coauthors)

Reviewer #1 (Remarks to the Author):

The authors studied the transport of ballistic Dirac Fermions in the topological insulators surface states of strained HgTe films where the surface has been engineered to manipulate the motion of charges. An antidot lattice that modifies the semi-classical trajectories of the electrons was defined, a signature of which can be observed in the magnetoresistance measurements where resonances appear each times that the cyclotron radius has some specific values related to the lattice constant of the square antidote lattice.

The experiments has been carried out in a thorough and proper way. Results are convincing and support the conclusions of this work. The authors are able to extract the energy dependence of the Fermi wave vector and the results are in very good agreement with previous results obtained with different experimental technics. The authors claim that the interest of their work lies in the fact that these experiments allow to directly access the value of the Fermi wave vector and that it demonstrates the ballistic nature of the motion of charges in HgTe 3D topological insulator.

I recommend this work for publication provided that the authors answer the concerns and remarks addressed below. The results are novel and the demonstration of the possibility to engineer topological surface states would have a clear impact for building electronic devices out of ballistic 3D topological insulators.

Remarks and concerns:

i) The authors show that their experiments directly probe the value of the Fermi wave vector. Nevertheless, the interest of this method with respect to other measurement technics that also probe k_F is not clearly explain in the manuscript. For instance, it remains unclear what would be the advantage over Shubnikov de Haas oscillations (SdHO) measurements (at low and high magnetic field to extract all densities according to some previous work of the authors) of either the conductance or the quantum capacitance and even over angle resolved photo emission spectroscopy experiments. Indeed, SdHO also probe k_F directly (and not the density of charge/state) as well as ARPES measurements. According to my understanding, the strongest conclusion of this work is the observation of ballistic motion and the manipulation of the charge orbitals based on the engineering of the surface state in strained HgTe thin film.

We fully agree with the Referee that the central and new finding of the present work is that we successfully imposed a periodic potential onto a TI system and that we observed a clear ballistic effect. To put this more into the focus we have omitted the following clause from the abstract: **“probing directly the Fermi wave vector of helical electron states on a single TI surface”**. One aspect of our manuscript is that we show that the antidot position gives the carrier density of the top surface if we use the relation $k_F = \sqrt{4\pi n_{top}}$ for non-spin degenerate Dirac electrons. Once this relation has been established one can use capacitance oscillations, the antidot resonance positions or low-field Shubnikov de Haas oscillations to determine the carrier density of the top layer. One advantage of this semi-classical method is that it allows significantly higher operation temperatures as compared to Shubni-

kov de Haas or capacitance oscillations, which rely on quantum interference. We added a corresponding sentence to the revised manuscript: **“Therefore, antidot resonances can be used to extract the carrier density of the top surface also at higher temperatures at which quantum oscillations are smeared out.”**

ii) To my understanding, measurements of the quantum capacitance give a direct access to the density of states and therefore to the electronic density. Nevertheless, the electronic density is given in this work by the magneto oscillations of the quantum capacitance which are related to k_F instead of the density. Therefore, it is not surprising that k_F is found to correspond to a non-degenerate surface state since both methods are directly related to k_F . In order to have a strong conclusion about the spin density, the authors should compare the k_F extracted from the measurement of the antidot lattice with the one obtained from direct measurement of the density, i.e. from quantum capacitance measurements. If this would not be possible, they should not claim that the comparison of the k_F between both methods is a proof of the non-degeneracy of the surface states.

While the value of the quantum capacitance is, under appropriate experimental conditions, directly connected to the density of states, the period of the quantum oscillations of the capacitance gives, as SdHO, a carrier density. We can use only the latter method to determine the carrier density as extracting the carrier density from the zero-field capacitance would be much too inaccurate (parasitic capacitances, unknown potential drops within the 80 nm thick HgTe film). In Kozlov et al. [Phys. Rev. Lett. **116**, 166802) we have “calibrated” the carrier densities extracted from capacitance oscillations against quantum Hall plateaus, so that we are confident that we have an independent carrier density. However, to address the Reviewers concern regarding this point we have replaced the word “confirm” in the following sentence by “is in line”: “The nearly perfect agreement between measured and calculated resonance position **is in line** with antidot resonances stemming from helical Dirac surface states which are spin-polarized, as sketched in Fig. 3b, and probe a single topological surface.” Further we have modified the last sentence of the manuscript which now reads: “The antidot resonances probed here reveal the Fermi wave vector, **being consistent** with Dirac surface states **which** are singly occupied in k-space.”

iii) For the 400nm, a correction accounting for the difference between the area of the antidot lattice and the total area would clearly strengthen the conclusion of this work. Three different lattice constants make the statistics strong enough as long as all of them are in good agreement with the theory. Even if the argument of the authors appears very reasonable, a quantitative estimation of the effect of the difference of areas would make their conclusions even more convincing.

The patterned area covers in case of the 408 nm sample only ~18% of the total capacitor area, the unpatterned region 82%. Unfortunately no reference area was on the sample with the 408 nm antidots so that we cannot directly compare carrier densities in patterned and unpatterned areas of this sample. Assuming that the carrier density in the antidot area is by ~20% lower than in the reference region (which is a typical value) and taking the ratio of the areas into account describes the position of the antidot resonances in Fig. 4b quite well. In the revised version we show in Fig.4b, besides the solid green line which was calculated by using the average carrier density derived directly from the capacitance oscillations, the corrected line which assumes a 20% lower density in the antidot area. We write: **“By assuming a 20% reduced carrier density in the antidot patterned areas compared to pristine ones, a plausible value for our system, we obtain a perfect match with experiment (green dashed line in Fig.4b)”** In the figure caption to Fig.4b we write: **“The dashed line for the 408 nm sample shows the calculated $B_1(n_s)$ trace with corrected carrier density (see text).”**

iv) Even if the three different lattice constants are very convincing, it would be worth to realize experiments on different lattices (for instance a triangular lattice or a lattice with two different lattice constant) in order to demonstrate the relevance of antidot lattices for the manipulation of ballistic Dirac Fermions. Nevertheless, this should not be a requirement for publication but a suggestion to make the conclusion of this work even more interesting.

We agree with the Referee that varying the symmetry of the antidot lattice and showing that the geometric resonances correspond to the modified commensurability conditions would be a further demonstration of the correctness of the antidot picture. So far we have concentrated only on simple square lattices with different periods so that we can presently show only data from square antidot lattices.

v) The authors did not discuss the nature of the perturbation induced by the antidot (weak or a strong perturbation). Strong or weak perturbation refers here either to dots which are completely depleted (Ensslin et al. Phys. Rev. B 41, 12307, Weiss et al. Phys. Rev. Lett. 66, 2790, Kang et al. Phys. Rev. Lett. 71, 3850) or to a situation where the potential varies only weakly in the dot (Weiss et al. EPL 8, 179, Lorke et al. Phys. Rev. B 44, 3447, Deng et al. Phys. Rev. Lett. 117, 096601). One might naively expect a weak perturbation but the authors did not discuss this point in the manuscript. The work Lorke et al. (Phys. Rev. B 44, 3447) which justifies the interpretation of the data as done in the manuscript, should be cited here. Even if I imagine that the answer is probably negative, a relevant question would be: is there any experimental way to differentiate both scenarios?

Although we do not know the exact microscopic origin of the artificial scattering potential we can conclude from experiment that we are very likely in the strong modulation limit. The resonances in Fig. 1d, e.g., occur when $2R_c/a = 1, 1.6$ and 2.28 , as pointed out in the figure caption of Fig.1. Also the data of the 600 nm sample, displayed in Fig. 3d (middle panel), show faint features which fully agree with this sequence. This is the characteristic sequence of resistance peaks in a strong antidot potential, demonstrated in Weiss et al. Phys. Rev. Lett. 66, 2790 (1991). Here, “strong” means that the potential specularly deflects all impinging electrons. The fact that the peak of the orbit around two antidots is shifted to higher fields as expected from the simple geometric consideration is known as a consequence of the soft (electrostatic) potential, in contrast to a hard wall potential [R. Fleischmann et al., PRL68, 1367 (1992)]. In other words: the sequence of peaks and their position is very similar to the ones observed in GaAs based antidot systems. In case of a weak modulation one would expect the sequence $2R_c/a = 1.25, 2.25$ and 3.25 , not consistent with the experimental observation here. While the situation is clear for the 800 nm and the 600 nm sample, as more than one resonance position is resolved in experiment and the magnetoresistance drops rapidly after the fundamental antidot resonance, one might argue that the single resonance peak in the 408 nm sample might stem from a weaker perturbation. The fact that the maximum occurs at $R_c/a = 0.5$ and not at $R_c/a = 0.625$ as expected for a weak periodic potential where a phase factor of $1/4$ needs to be taken into account (although it has been pointed out by Lorke et al. Phys. Rev. B 44, 3447, that the phase factor in 2D depends on the exact shape of the potential), the height of the zero-field resistance, and the fact that no higher order oscillations appear (expected for a weak periodic potential when the ratio of mean free path and period decreases) points also here to a strong potential. In the revised manuscript we have added in Fig.3d arrows to the position where one expects orbits around two and four antidots. We also add the bold face text to the caption of Fig.1d: “The corresponding arrow posi-

tions, **characteristic for a strong antidot potential**, have been calculated using $B_1 = \hbar k_F / (0.5ea)$, $B_2 = \hbar k_F / (0.8ea)$, and $B_4 = \hbar k_F / (1.14ea)$, with $k_F = \sqrt{4\pi n_s^{top}}$

We added further a sentence describing in more detail the antidot potential we believe is present in our system: **“From the similarity to antidot resonances in GaAs based 2DES we conclude that the antidot potential has the diameter of the etched holes and drops precipitously but soft towards the region between the antidots.”**

We also included the reference Lorke et al. Phys. Rev. B **44**, 3447.

vi) The inset in the figure 4.b is not clear. In general, it would be good to have the title of the different axis (inset included)

In Fig. 4b, the axes are, as far as we can see, correctly labelled in the graph. The Referee probably refers to Fig. 4a. In the revised version we have included the titles of the axes in the graph.

vii) In the introduction, it is very difficult to make the link between Majorana, spintronic and the present experiment. A more detailed description would be helpful for the reader to understand the interest of this work regarding those two fields.

The link was meant very general: For probing Majorana bound states one can resort to 1D structures or placing patterned superconductors and/or ferromagnets on top of TIs. In any case one patterns the TI on a 100 nm scale or smaller. And the present work is on the effect of patterning TIs on the 100 nm scale. In the revised version we had to adapt the style of the abstract to the one of Nature Communications with no references in the abstract. Therefore we have rewritten a bit the abstract and refrain from introducing Majorana bound states or spintronics issues and a broader discussion of it as it would detract from the main topic. We therefore have replaced the first sentence of the abstract and of the manuscript.

viii) Do you have any explanation for the non-monotonous change of the resistance between 400 nm, 600 nm and 800 nm samples for the resistance? Is it a due to the geometry of the sample? More generally, the exact geometry of the Hall bars are not indicated (length, width, width of the contact, number of contact, etc...). The schematic in the Fig. 1 is a nice drawing that makes clear how the samples are measured but it is not sufficient to get all useful information.

In Fig. 3b which shows magnetoresistance of the three devices at the same carrier density, the 600 nm sample has a lowest zero field resistivity (63 Ω) while one would have expected that the resistivity is in between the one of the 800 nm (99 Ω) and the 408 nm (212 Ω) antidot lattice. We have no definite answer but speculate that this has to do with the larger thermal load seen by this sample. Due to problems during atomic layer deposition the sample was (after patterning) stored a longer time at 80°C in the ALD apparatus. One scenario could be that this procedure steepens the soft antidot potential, the antidot lattice becomes more open and the resistance drops. The fact that the peak corresponding to the orbit around 2 antidots is for the 600 nm sample in Fig.3d closer at the pure geometric condition $2R_c/a = 1.6$ than for the 800nm sample points in this direction. Anyhow, as the sequence of peaks is essentially unchanged between 600 and 800 nm sample we conclude that this does not affect the strong antidot potential. In the revised manuscript we write in the figure caption of Fig. 3d: **“The lower zero field resistance of the 600 nm sample we ascribe to a longer exposure to 80°C during insulator deposition.”**

In the Methods section we have added a more detailed description of the Hall bar which reads: **“We used conventional Hall bar geometries having a width of 50 μm and a length of 650 μm with 4 potential probes of 5 μm width on each side, separated by either 100 μm or 250 μm . The resistivity $\rho_{xx}(B)$ was extracted from areas of size 100 μm x 50 μm or 250 μm x 50 μm , respectively.”**

Reviewer #2 (Remarks to the Author):

The authors have created submicron periodic potentials onto the top surface of strained mercury telluride and they have observed pronounced geometric resistance resonances when the Fermi wave vector of the TI matches the size of the antidot arrays.

I think the present work is of high quality and it might be of great interest to others in the community and the wider field. The experimental data supports the claim of ballistic effects in 3D-TIs and there is enough evidence from the 3 different samples mentioned in the manuscript. Therefore, I would strongly recommend the paper for publication in Nature Communications. I only have minor comments.

1. Experiments and theories on antidot superlattices in graphene has been performed before, and they should be cited in this manuscript. Such examples: Pedersen et al. and T. Shen et al.

In the revised version we now add references to antidots in graphene including the ones of Pedersen et al. and Shen et al.. We added the following sentence with references to the manuscript: **“Similar effects were predicted and observed in graphene, also a system with Dirac like dispersion⁹⁻¹¹”**

2. A brief explanation why the capacitance measurements only gives the top-surface carrier density should be included in the paper, since it is important for the main conclusions in the paper.

The fact that the top layer dominates is most easily seen in case of a fully metallic top surface layer which screens the entire electric field from the metallic top gate. In this case the carrier density of the bottom layer does not change as a function of gate voltage, only the one of the top layer. Here we have a two-dimensional electron gas which does not perfectly screen the field, but the top layer still dominates the capacitance. Experimentally this has been shown in Kozlov et al. Phys. Rev. Lett. **116**, 166802 (2016). In the revised manuscript we write now: **“In capacitance experiments the top layer dominates, as it screens, to a large extent, the electric field from the gate.”**

Reviewer #3 (Remarks to the Author):

I reviewed the paper “Ballistic geometric resistance in a single surface of a topological insulator” by Maier et al. The paper presents an experiment in which a periodic array of anti dots is patterned on the surface of a 3DTI by etching. The authors present a set of transport measurements in which resonances of the magnetoresistance are identified and from which the Fermi wave vector is extracted.

The setup is borrowed from an old set of experiments done on 2DEGs that targeted, for example, the physics of the half filled Landau level and from which one could extract the Fermi wave vector of composite fermions. Of course there, actual holes were punched into the 2DEG. Here, the authors use the unique property of the surface of the TI, which is that it sticks to any geometry and cannot be interrupted by non-magnetic perturbations. Hence, the surface state follows the bumps and curves of

the etched device, unlike in the standard 2DEG.

The particles seem to avoid the antidots in the periodic array due to an electrostatic potential, although in principle the surface state does not terminate there. The authors state that the precise mechanism by which this repulsion is generated is not known but it is likely to form due to different doping level created in these regions. The data clearly shows resonances in the expected positions so in this sense it seems to agree with the mechanism described in the manuscript.

Overall I find the manuscript of interest and the data convincing. Patterning of TI materials is an outstanding problem and has proven challenging in the past. It did not work well with magnets, for example. I think this design could prove useful if adapted in other cases as well, such for the anomalous Hall insulator, in which the transition into the topological phase crucially depends on the thickness of the TI. Therefore a proof of concept for patterning in TI is interesting and important. I am generally positive about this paper being published in Nature Communications. Before it is published I would like the authors to consider some points for discussion.

I think the authors should discuss in more detail how the electrostatic potential is formed and affects the structure and the effective length scales. If the particles avoid the antidot due to electrostatic repulsion, I would expect that the potential is not hard-wall like but soft and smooth in contrast with the drilled holes in 2DEGs. Is there an indication for that in any of the measurements? What is the effective size of an antidot? This is not a point like object as is seen clearly from fig1, its size is comparable to the distance between antidots.

Antidots generally have no hard wall potential (R. Fleischmann et al., PRL68, 1367 (1992) – in GaAs based antidots it is the surface charges localized at the perimeter of the antidot which give rise to a soft electrostatic potential. The antidots in our experiment have a diameter between 170 nm and 190 nm (**this info is now contained in the methods section**), as deduced from electron micrographs. Signatures of a soft potential become clearly seen in experiment when circular cyclotron orbits get deformed by the periodic potential, an effect which modifies the commensurability condition for the higher order resonances. For the 600 nm and the 800 nm lattice this effect is generally small as the peak around four antidots occurs largely at the magnetic field position expected for circular orbits. The peak corresponding to an orbit around two antidots, however, is especially prone to deformation (Fleischmann et al.). Indeed a small shift is seen in Fig. 1d (and also in Fig. 3d for the 600nm and 800nm sample) where the peak corresponding to an orbit around two antidots is slightly shifted to higher fields compared to the calculated one occurring at $2R_c/a = 1.6$ (arrow position). This is a signature that we do not have a hard wall potential. For the 408 nm sample the peak around two antidots is no longer resolved indicating that the decay length of the repulsive antidot potential is of the order of 100 nm and cuts off orbits around 2 and 4 antidots. Although we have no microscopic picture (in addition to local doping also a modification of strain at the etched position in this strained material can contribute to the antidot potential) the features seen in experiment are very similar to the ones observed in GaAs based antidot lattices so that we assume that the diameter of the antidot potential is given by the diameter of the etched holes and decays with a characteristic length of a few ten nanometers. In the revised manuscript we write: **“From the similarity to antidot resonances in GaAs based 2DES we conclude that the antidot potential has the diameter of the etched holes and drops precipitously but soft towards the region between the antidots.”**

In this context, I would also like to point out a related experimental study (see <https://arxiv.org/abs/1207.4468>) where periodic buckling was introduced in Bi₂Te₃ and resulted in

a one-dimensional periodic potential. Among other things, in this paper, the authors discuss the effects of doping vs potential ripples from lattice deformations. It would be good if the authors could compare and contrast the physics presented in this paper with the one that emerges from the antidot array.

We thank the Referee for drawing our attention to this publication. There might indeed be a close connection as we are working with a strained material: HgTe is grown on CdTe which has a larger lattice constant. The topological nature of HgTe stems from a gap in the Γ_8 bands which opens due to the tensile strain. In this gap the gapless surface states form. By etching holes into the top layer the strain pattern is likely periodically modulated and contributes to the antidot potential. In our revised version we write: **“By etching holes into the top layer the strain pattern is likely periodically modulated and contributes to the antidot potential in a similar way as ripples affect the band structure in Bi_2Te_3 ”**²⁰

Finally, I would like to understand if the Berry phase of the Dirac particles could be directly probed by this experiment. Since we are discussing transport in the semiclassical regime, the semiclassical equations of motion should in principle contain such a contribution.

Electron spins on a cyclotron orbit in a TI acquire a Berry phase. This phase can be observed in Shubnikov de Haas oscillations since Landau quantization can be viewed semi-classically as interference of an electron's wave function along its cyclotron orbit. The Berry phase gets, e.g., manifested by a phase shift of about one half an oscillation index when one plots the position of an oscillation minimum of Shubnikov de Haas oscillations on a $1/B$ scale vs. the oscillation index (see e.g. Y. Ando, Journal of the Physical Society of Japan 82 (2013) 102001). Also in our antidot lattice electrons on a commensurate orbit should acquire a Berry phase which comes to light if the motion gets quantized and quantum oscillations appear as observed and described in Weiss et al., PRL **70**, 4118 (1993). What we probe here by means of geometric resonance is only the classical trajectory of the orbits without phase information. In order to resolve quantum oscillations superimposed on the classical commensurability peak one needs to go to mK temperatures. This we have not done yet.

REVIEWERS' COMMENTS:

Reviewer #1 (Remarks to the Author):

I am generally satisfied (and/or convinced) with the answers of the authors except for the point ii) raised in my first report. I try below to explain in details my concern about this –minor– conclusion of the present work.

I agree (and agreed) about the original formulation in the manuscript regarding the terms “confirms” (original sentence: “The nearly perfect agreement between measured and calculated resonance position confirms that the antidot resonances stem from helical Dirac surface states which are spin-polarized, as sketched in Fig. 3b, and probe a single topological surface”) and “showing directly” (original sentence: “The antidot resonances probed here reveal the Fermi wave vector, showing directly that the Dirac surface states are singly occupied in k-space.”). Nevertheless, I do not agree with the conclusion of the authors about the confirmation or the evidence of probing a non-degenerated Fermi circle (second part of the original sentences above).

Indeed, to my knowledge, SdHO as well as quantum oscillations of the quantum capacitance (QOQC) do not give directly access to the carrier density as claimed by the authors (contrary to the direct measurements of the quantum capacitance) rather to the cross-sectional area of the Fermi surface in a plane normal to the B field (see for instance Kittel’s book). This quantity is related to the value of k_F . SdHO (or the van Alphen effect) are therefore commonly used to determine the Fermi surface of different metals. Hence, this method gives the carrier density providing that we know the relation between carrier density and k_F . It requires that the spin degeneracy of the band structure is known. This is for instance the case in measurements of the carrier density of a 2D electron gas in a GaAs heterostructure or of a graphene sheet. Finally, the fact that the results of the antidot resonances are in very good agreement with the QOQC or with the SdHO is a very strong indication that all those methods probe the value of k_F in a coherent way but it does not show “directly that the Dirac surface states are singly occupied in k-space”. To do so, the authors would have to compare the results between a method that gives access to k_F directly (antidot resonances, SdHO, QOQC,...) and a method that gives access to the carrier density directly (quantum capacitance, Hall effect – very difficult here –, ...)

Providing that this point can be clarified in a new version, I still believe that this work should be suitable for publication in Nature Communication.

Reviewer #3 (Remarks to the Author):

I review the paper "Ballistic geometric resistance resonances in a single surface of a topological insulator" for the second time, along with the reports and response letter presented by the authors. I find the authors' response comprehensive and satisfactory, taking into account the criticism put forward by the referees. I believe that the paper is ready to be published in Nature Communications.

Response to Referees

We thank Referee 1 for insisting on this point which allows us to clarify this point in the manuscript. We hope that our answer below and the changes in the manuscript clear up this issue.

Dieter Weiss (on behalf of all coauthors)

Reviewer #1 (Remarks to the Author):

I am generally satisfied (and/or convinced) with the answers of the authors except for the point ii) raised in my first report. I try below to explain in details my concern about this –minor– conclusion of the present work.

I agree (and agreed) about the original formulation in the manuscript regarding the terms “confirms” (original sentence: “The nearly perfect agreement between measured and calculated resonance position confirms that the antidot resonances stem from helical Dirac surface states which are spin-polarized, as sketched in Fig. 3b, and probe a single topological surface”) and “showing directly” (original sentence: “The antidot resonances probed here reveal the Fermi wave vector, showing directly that the Dirac surface states are singly occupied in k-space.”). Nevertheless, I do not agree with the conclusion of the authors about the confirmation or the evidence of probing a non-degenerated Fermi circle (second part of the original sentences above).

Indeed, to my knowledge, SdHO as well as quantum oscillations of the quantum capacitance (QOQC) do not give directly access to the carrier density as claimed by the authors (contrary to the direct measurements of the quantum capacitance) rather to the cross-sectional area of the Fermi surface in a plane normal to the B field (see for instance Kittel’s book). This quantity is related to the value of k_F . SdHO (or the van Alphen effect) are therefore commonly used to determine the Fermi surface of different metals. Hence, this method gives the carrier density providing that we know the relation between carrier density and k_F . It requires that the spin degeneracy of the band structure is known. This is for instance the case in measurements of the carrier density of a 2D electron gas in a GaAs heterostructure or of a graphene sheet. Finally, the fact that the results of the antidot resonances are in very good agreement with the QOQC or with the SdHO is a very strong indication that all those methods probe the value of k_F in a coherent way but it does not show “directly that the Dirac surface states are singly occupied in k-space”. To do so, the authors would have to compare the results between a method that gives access to k_F directly (antidot resonances, SdHO, QOQC,...) and a method that gives access to the carrier density directly (quantum capacitance, Hall effect – very difficult here –, ...)

Providing that this point can be clarified in a new version, I still believe that this work should be suitable for publication in Nature Communication.

Our reply: We fully agree with the Referee that oscillations of the quantum capacitance (QOQC) measure the Fermi surface and that one needs to know the spin degeneracy of the band structure to convert the $1/B$ periodicity of the oscillations into a carrier density. The central point, which has not been sufficiently pointed out in our manuscript, is that we have shown before [Kozlov et al., Phys. Rev. Lett. **116**, 166802]) that QOQC probe the top surface only and that the corresponding spin-degeneracy factor is $g_s = 1$. Although we cannot extract the carrier density directly from the abso-

lute value of the quantum capacitance (too inaccurate) or from the Hall effect (probes top and bottom layer together and provides total carrier density), as suggested by the Referee, we can reliably extract the carrier density from the QOQC using $g_s=1$. Given that we are confident that our statement “The antidot resonances probed here reveal the Fermi wave vector, being consistent with Dirac surface states which are singly occupied in k-space” is correct. In order to clarify that we know the degeneracy factor g_s which relates the period of the QC oscillations to the carrier density we added the following sentence to the manuscript: “Here, we have used that the spin-degeneracy factor g_s , utilized to extract the carrier density from the period of the capacitance oscillations, is 1, i.e. the Landau levels are not spin degenerate [Kozlov et al. Phys. Rev. Lett. **116**, 166802].” In the cited reference we explicitly stated that the quantum capacitance oscillations stem from non-degenerate LLs.